# Molecular Markers Associated with Agro-Physiological Traits under Terminal Drought Conditions in Bread Wheat

**DOI:** 10.3390/ijms21093156

**Published:** 2020-04-30

**Authors:** Sajid Shokat, Deepmala Sehgal, Prashant Vikram, Fulai Liu, Sukhwinder Singh

**Affiliations:** 1Department of Plant and Environmental Sciences, University of Copenhagen, Højbakkegård Allé 13, 2630 Taastrup, Denmark; fl@plen.ku.dk; 2Wheat Breeding Group, Plant Breeding and Genetics Division, Nuclear Institute for Agriculture and Biology, Faisalabad 38000, Pakistan; 3International Maize and Wheat Improvement Centre (CIMMYT) km, 45, Carretera Mex-Veracruz, El-Batan, Texcoco CP 56237, Mexico; D.Sehgal@cgiar.org; 4International Potato Center, NASC Complex, Pusa, New Delhi 110012, India; pvikramseedwheat@gmail.com; 5Geneshifters, 222 Mary Jena Lane, Pullman, WA 99163, USA

**Keywords:** bread wheat, haplotype blocks, GWAS, grain yield, kernel abortion

## Abstract

Terminal drought stress poses a big challenge to sustain wheat grain production in rain-fed environments. This study aimed to utilize the genetically diverse pre-breeding lines for identification of genomic regions associated with agro-physiological traits at terminal stage drought stress in wheat. A total of 339 pre-breeding lines panel derived from three-way crosses of ‘exotics × elite × elite’ lines were evaluated in field conditions at Obregon, Mexico for two years under well irrigated as well as drought stress environments. Drought stress was imposed at flowering by skipping the irrigations at pre and post anthesis stage. Results revealed that drought significantly reduced grain yield (Y), spike length (SL), number of grains spikes^−1^ (NGS) and thousand kernel weight (TKW), while kernel abortion (KA) was increased. Population structure analysis in this panel uncovered three sub-populations. Genome wide linkage disequilibrium (LD) decay was observed at 2.5 centimorgan (cM). The haplotypes-based genome wide association study (GWAS) identified significant associations of Y, SL, and TKW on three chromosomes; 4A (HB10.7), 2D (HB6.10) and 3B (HB8.12), respectively. Likewise, associations on chromosomes 6B (HB17.1) and 3A (HB7.11) were found for NGS while on chromosome 3A (HB7.12) for KA. The genomic analysis information generated in the study can be efficiently utilized to improve Y and/or related parameters under terminal stage drought stress through marker-assisted breeding.

## 1. Introduction

Wheat (*Triticum aestivum* L.) is grown in more than 85 countries with about 2.1 million km^2^ total harvested area and contributes to about 20% of the total dietary calories and proteins worldwide [1]. It is cultivated in environments ranging from very favorable ones in Western Europe to severely stressed ones in parts of Asia, Africa, and Australia, thereby facing various biotic and abiotic stresses.

Of all the abiotic stresses curtailing wheat productivity, drought has the most detrimental effects in rainfed environments [2]. Terminal drought stress i.e., drought stress at the time of flowering reduces grain yield through ceased growth of spike and reduction in grain number and grain weight [3,4,5,6]. Ultimately, grain yield under drought stress is reduced drastically. Uneven patterns of rainfalls and more frequent spells of drought with ongoing climate change will become more frequent in the future and will act as fatal threats for wheat production. There is a pressing need to develop wheat varieties that produce higher yield with less water. 

Conventionally, genetic gain has been achieved following traditional breeding methods, in which, success largely depends on genetic potential of the breeding pool and selection efficiency. Traditional breeding methods have achieved tremendous success in the last century [7] by pushing the annual genetic gain in wheat through the use of desirable genetic variations within primary gene pools. Genetic resources such as landraces and wild relative collections, maintained in gene banks, are a reservoir of unexplored beneficial alleles. Worldwide wheat breeders have access to up to 800,000 accessions, many of which show adaptation to different abiotic stresses, for example, Creole wheat landraces that were introduced to Mexico from Europe around five centuries ago [8]. Breeders are usually reluctant to introgress these rare unexplored alleles into wheat cultivars because of the challenges involved in identifying useful, novel diversity and transferring into well-adapted elite cultivars with minimum linkage drag. Developing an intermediate germplasm between adapted and non-adapted through exotic × elite crosses is one of the alternatives to address this issue.

Pre-breeding plays a key role in minimizing the barrier in mobilizing novel under-utilized genetic variation into breeding programs through introducing less undesired, linked genes. It refers to the development of bridging, semi-finished, or intermediate germplasm having introgressions from exotic or unimproved germplasm [9]. The Seeds of Discovery (SeeD) project was launched at CIMMYT to create new pre-breeding germplasm in order to enhance the use of genetic resources in breeding programs [10]. Within the framework of project, a panel of pre-breeding lines (PBLs) has been generated by three-way (exotic/elite1//elite2) crosses of hundreds of exotics (landraces and synthetics) with elites [11]. These PBLs have been distributed to different international partners for multi environment trials for the release of varieties and for novel gene discoveries for complex traits [12]. Next generation wheat varietal improvement can be achieved through utilizing potential of germplasm banks and advanced genomic technologies. Specifically, with the use of genomics approaches, breeders are now able to enhance breeding efficiency while simultaneously addressing bottlenecks of breeding.

With the upsurge in genotypic data coming from different high-density platform, a genome wide association study (GWAS) approach has become a common approach for dissection of traits. GWAS studies have been reported for various traits in wheat, including traits with complex genetic architectures such as grain yield, yield components, and physiological traits under multiple environments [13,14,15,16,17,18,19,20]. GWAS aided by a haplotype based mapping approach has proven to be a powerful method for complex traits [12,21,22]. Identification of genomic regions linked with traits of breeder’s interests including grain yield under drought can help breeders in different ways. These improve the enrichment of breeding germplasm pools with positive alleles as well as direct introgression through marker assisted selection.

The current study was conducted on a subset of 339 PBLs, which were developed at CIMMYT under SeeD project and evaluated for Y under irrigated, heat, and drought stress conditions [11]. Current study was focused on Y related parameters under terminal stage drought stress condition. The objective was to identify the genomic regions linked with Y and related traits through marker-traits associations (MTAs). This mapping panel was grown under irrigated and terminal stage drought stress environments to quantify the yield losses and maintenance of yield potential in different PBLs over the period of two years. GWAS was performed through single marker and haplotypes-based mapping while candidate genes linked with the genomic associations were reported. Information generated in this study can be used for genetic enhancement of breeding germplasm pools for Y and related traits under terminal drought stress conditions as well as improved understanding of genetic basis of traits investigated.

## 2. Results

### 2.1. Analysis of Variance and Heritability

Significant differences for grain yield (Y), canopy temperature depression (CTD) days to heading (DTH), and days to maturity (DTM) were observed for the years 2016 and 2018 (*p* < 0.001) and the lowest grain yield was recorded in 2016 (Table 1 and; Figure 1a and Appendix A). However, DTH and DTM were significantly reduced during 2018 (Appendix A). Drought significantly reduced Y, spike length (SL) and number of grains spike^−1^ (NGS) (Table 1 and; Figure 1a, Figure 2a and Figure 3a), normalized difference in vegetative index (NDVI), CTD (Table 1 and; Appendix A), plant height (PH), DTH, and DTM (Appendix A) in comparison to irrigated conditions (*p* < 0.001). In contrast, kernel abortion (KA) was significantly increased (Figure 4a) and thousand kernel weight (TKW) was significantly reduced under drought conditions (Table 1 and Figure 5a). All genotypes differed significantly for Y, SL, TKW, PH, and DTM. Range of Y reduction was 25.7% to 46.43% in pre-breeding lines designated as GID-7645073 and GID-7641264, respectively (Figure 6a). The interaction of genotypexyear and yearxtreatment were significant for Y (Table 1). Maximum reduction, 19.2% in SL, was recorded in GID-7645435, while 78 genotypes were found with no reduction in SL in comparison to irrigated conditions (Figure 6b). Highest 12.1 cm SL under drought was recorded in GID-7640992. Highest reduction 23.5% for NGS was recorded in GID-7644595 and out 339 genotypes, and 18 were found with no reduction in NGS (Figure 6c). GID-7643397 was most sensitive with 11.6% reduction in KA. Maximum reduction 25.9% for TKW was recorded in GID-7644025 and 32 genotypes maintained TKW without any reduction (Figure 6d). Highest average two years TKW (58.77 g) under irrigated conditions was recorded in GID-7645143. Interaction between genotypextreatment was significant for TKW and NDVI. while interactions between genotypesxyear and treatmentxyear were also significant for PH and DTM (Table 1). 

Broad sense heritability (h_bs_) was 0.61 and 0.65 for Y, 0.80 and 0.72 for SL, 0.72 and 0.6 for NGS, 0.67 and 0.84 for TKW, 0.25 and 0.68 for NDVI, 0.13 and 0.22 for CTD, 0.81 and 0.71 for PH, 0.77 to 0.08 for DTH and 0.85 and 0.82 for DTM for irrigated and drought conditions, respectively. Very low (0.17) h_bs_ was found for KA under drought conditions (Table 1).

### 2.2. GWAS Analysis in Diverse Pre-Breeding Germplasm

#### 2.2.1. Physiological Traits

Individually, GLM and MLM revealed a number of associations; however, associations explained by both were further taken to explain the traits. GLM and MLM analysis revealed significant (*p* < 0.001) associations of NDVI on different chromosomes; however, common associations at the same chromosome were selected for irrigated and drought conditions, where significant and common associations were found on chromosome 1B and 3A for year 2018 (Appendix A). For CTD, associations were calculated on chromosome 4B and 7B for irrigated and drought conditions, respectively (Appendix A). Association of CTD was also significant at 4A chromosome under drought conditions. 

#### 2.2.2. Plant Height and Earliness 

GLM and MLM of GWAS indicated a significant association (*p* < 0.001) of PH at chromosome 1B and 2A under irrigated and drought conditions for the year 2016, respectively, while common association was found at chromosome 7A for the year 2018 (Appendix A). No common association for DTH was recorded; however, associations on chromosomes 3A, 2A, 1B, and 7A were recorded for irrigated and drought conditions of 2016 and irrigated and drought conditions of 2018, respectively (Appendix A). Moreover, on chromosome 3A, favorable allele TA of HB7.17 showed earliness for DTH in comparison to CC allele under irrigated conditions (Appendix A). Like DTH, no common association for DTM was recorded under irrigated conditions, where association on chromosomes 5B and 5A was recorded for the years 2016 and 2018, respectively. However, association on the same chromosome (6A) was recorded under drought conditions for both years (Appendix A). Furthermore, HB2.5 on chromosome 1B with favorable allele AT showed earliness for DTM in comparison to other alleles (Appendix A). 

#### 2.2.3. Grain Yield (Y) and Yield Related Traits

For Y, common significant associations (*p* < 0.001) were identified on chromosomes 4A and 6B in both conditions and years (Figure 1b–e and Table 2). Haplotype based analysis also identified HB10.1 on chromosome 4A to be associated with Y with haplotype TCG as a favorable allele as compared to CTC for the year 2018 in both conditions (Figure 1f). Likewise, another association on chromosome 4A, HB10.6, with favorable allele AC showed higher Y in comparison to CT for 2016 (Appendix A) in both conditions (Figure 1g,h). Wheat genotypes showing less average reduction in Y were also showing favorable alleles (AC) for HB10.7 while genotypes showing highest average reduction in Y were confirming un-favorable allele (CT) for HB10.7 under drought (Appendix A). For SL, GWAS analysis indicated significant (*p* < 0.001) associations on chromosomes 2D and3D under drought and irrigated treatments in 2018 (Figure 2b,c and Table 2). Chromosome 2D was further analyzed and HB6.10 with favorable allele TC showed higher spike length compared to other alleles (TG or CC) under drought and irrigated conditions (Figure 2d). GLM and MLM identified significant associations of NGS on chromosomes 6B and 3A both under irrigated and drought conditions for the year 2018 (Figure 3b,c and Table 2). Haplotype CC of HB17.1 on chromosome 6B showed significantly higher NGS for 2018 under both treatments (Figure 3d). Similarly, haplotype CT on chromosome 3A (HB7.11) is responsible for higher NGS under drought in comparison to allele TT or CG (Figure 3e). Strong associations for KA were identified on chromosomes 3A and 3B. Haplotype CGGTC from HB7.12 showed lower KA compared to TTCGT under drought conditions (Figure 4b,c and Table 2). For TKW, common association was identified on chromosome 3B for both years. Similarly, significant association was also found at 4A under drought conditions for the year 2018 (Figure 5b–d and Table 2). The haplotype-based analysis indicates that haplotype TAGGCT (HB8.12) is favorable in comparison to TAGGCC or CGATC under irrigated condition of both the years (Figure 5e). 

### 2.3. Multiple Quantitative Traits Loci (QTL’s) at the Same Chromosome

Single marker trait analysis revealed significant associations of NDVI, DTH, NGS, and KA at chromosome 3A under both irrigated and drought conditions. TKW and KA were also significantly associated at chromosome 3B under drought conditions (Table 2 and Appendix A). Significant associations of CTD, TKW and Y were reported at chromosome 4A. Moreover, maker–trait association of DTH, NGS, and Y was significantly associated on chromosome 6B. 

#### Haplotype Blocks at Same Chromosomes

Genomic regions were identified in the haplotype block trait analysis. Two adjacent HBs on chromosome 3A, HB7.11, and HB7.12 were found to be significantly associated with NGS and KA, respectively. Similarly, three HBs, HB10.1, HB10.6, and HB10.7 were found to be associated with Y under a terminal stage drought stress though with a low significance level (Table 3 and Figure 7) 

## 3. Discussion

The use of haplotypes approach maximizes linkage disequilibrium between markers and QTLs as compared to SNPs. For traits with complex genetic architecture, use of multi-allelic haplotypes has significantly improved the power and robustness of GWAS in crops ([23,24,25]. In wheat too, recent studies have shown better resolving power of haplotypes in GWAS than SNPs [12,21,22,26]. We conducted both single marker and haplotypes based GWAS to dissect QTLs for drought tolerance in a panel of pre-breeding lines derived by three-way crosses of exotics with CIMMYT’s best 25 elites. 

The PBL panel investigated in the present study has been analyzed for population structure and LD decay in previous study which showed a moderate population structure and an LD decay of 2.5 cM genome wide [12]. This indicates its higher genetic diversity as compared to various other wheat panels used in previous studies reporting from 5 to 10 cM or even slower LD decay [13,17,27,28,29,30,31,32]. These results are expected since the panel has been drawn from a large set of pre-breeding lines developed by crossings among exotics and elites [11]. Each pre-breeding line acquired approximately 25% of the exotic genome and 75% of the elite genome at an early stage thus allowing recombination between exotic and elite genomes. Furthermore, many of the agronomical and physiological traits investigated on the panel including DTM, SL, NGS, TKW, and Y showed high heritability both under irrigated and drought conditions. Shokat et al. [33] reported that plant traits exhibiting higher heritability and genetic advance can directly be selected for crop improvement. 

Both single marker and haplotype-based GWAS revealed the association of Y on chromosome 4A in both the years under irrigated and drought conditions (Figure 1b–e, Table 2). Though with a low significance level, association of three HBs for Y in the same genomic region clearly indicates its importance for Y under terminal sage drought (Table 3, Figure 7). A previous report of Singh et al. [11] also emphasized the importance of this region on Y under drought. Population used in this study was a randomly selected subset of what was used by Singh et al. [11]. Results on chromosome 4A in this study are in concomitance with the previous study. The significance level for HB association with Y was low in this study as compared to what was in Singh et al. [11], most likely due to change in the population structure. In addition, the level of drought stress in 2016 was not as high to reduce yields under stress by 30% even. These are the most probable reasons for low level of significance. Association of three chromosome 4A HBs (in which two are consecutive) in the same genomic region for Y under drought stress and concomitance of results with that of Singh et al. [11] clearly signifies the relevance of this genomic region for Y under different moisture regimes. Interestingly, the association of this genomic region with CTD indicates that the Y increase under drought was likely due to cooler canopies under drought stress environments. Results therefore suggested that the 4A genomic region is drought responsive. Previous studies have also reported drought tolerance QTL on 4A in bread wheat [11,17,29,34,35] with percentage variation explained as large as 41% in some studies [36]. The present and previous studies therefore suggest an important role of 4A chromosome in drought adaptation in wheat. Three haplotypes blocks, i.e., HB10.1 (TCG), HB10.6 (AC) and HB10.7 (AC), within 10.76 to 32.4 cM, resulted in significantly higher yield across years (Figure 1f–h). These haplotypes were not found significant for DTH, confirming that the Y increase was not because of drought escape. 

SL is an important yield contributing trait and longer compact spikes are one of the attractive plant traits to obtain higher Y in wheat [37]. Mwadzingeni et al. [38] reported a positive correlation of SL with grain yield under drought stress. These results indicate that higher SL can contribute towards the optimum yield under drought. In the present study, SL had high heritability and it showed significant associations on chromosome 2D (HB6.10) positioning between 142.74 cM to 147.6 cM under irrigated and drought conditions (Figure 2b,c), where allele TC was linked with higher SL. Previously QTLs for SL were reported by Chai et al. [39] on the short arm of chromosome 2D. Likewise, Zhai et al. [40] reported QTL related to spike morphology on short arm of 2D. 

Limited availability of water at the time of anthesis reduces pollen viability [41] and seed setting which increases kernel abortion (KA). Maintenance of high number of grains spike^−1^ (NGS) under drought is one of the important challenges in *T. aestivum* to obtain ideal yield under drought condition. This trait was found positively correlated with grain yield under drought and irrigated conditions [6,42]. We found higher heritability and significant association of NGS with chromosome 6B and 3A both under irrigated and drought conditions and trait KA was also associated with chromosome 3A (Figure 3b,c and Figure 4b). 

In this study, KA showed a strong association with grain yield related parameters, especially grain number, indicating the cause of grain yield reduction under terminal stage drought stress.

Likewise, NDVI and DTH were also associated with chromosome 3A. In line of these, Li et al. [22] reported the significant association of NGS at chromosome 3A. Likewise, Pradhan et al. [43] also reported the marker traits association of NGS on chromosome 3A explaining 12–16% of variability. Other studies indicate that chromosome 6B is the third major chromosome of wheat covering more than 5% of wheat diversity with more 2000 genes [44]. Nadolska-Orczyk et al. [45] reported the presence of a genes controlling supernumerary spikelets trait on this chromosome. In the present study, HB 17.1 (TC) and HB7.11 (CT) were found to be associated with higher NGS under drought positioning at 2.64 to 6.34 cM and 56.45 to 56.5 cM, respectively (Figure 3e,f). HB7.12 with allele CGGTC responsible for lower KA was present at 111.6 to 115.9 cM (Figure 4c). HB7.11 and HB7.12 are closely placed indicating that NGS and KA may be linked together. 

Thousand kernel weight (TKW) is one of the main contributors of grain yield [46] and less reduction in TKW enables plants to maintain ideal yield under drought conditions [47]. Liu et al. [48] reported a positive association of TKW with grain yield under irrigated and drought conditions indicating that plant genotypes having higher TKW under irrigated conditions often have a chance to maintain higher TKW under drought conditions. We found significant association of TKW at chromosome 3B (Figure 5b,c) and haplotype HB8.12 with allele TAGGCT was giving higher TKW under irrigated conditions in both years in contrast to other alleles and positioning between 49.77 to 53.18 cM (Figure 5d). Lehnert et al. [49] also reported the association of TKW at chromosome 3B under irrigated conditions. Luján Basile et al. [50] reported the presence of one haplotype block for TKW at chromosome 3B. Rustenholz et al. [51] mentioned that 3000 genes are located on this chromosome indicating the richness and diversity of information on this chromosome. Likewise, among six of the corresponding markers, 1,082,914 and 1,010,250 (Appendix A) were found to have their presence in bread wheat. The sequence of 1,010,250 is also reported in Chinese Spring of bread wheat by Ogihara et al. [52] but their function is unknown in bread wheat.

## 4. Materials and Methods

### 4.1. Plant Material

Three hundred and thirty-nine pre-breeding lines derived from the three-ways top-crosses and their performance was evaluated under terminal stage drought stress environments at CIMMYT’s experimental station (CENEB), Obregon, Mexico. These 339 pre breeding lines (PBLs) were selected as a subset of the advanced PBLs explained in a previous report of Singh et al. [11] for phenotypic screening under drought stress environments. The germplasm material advanced to TC1F7 stage at CIMMYT was used in our experiments. These 339 PBLs were derived from crosses of 25 elite genotypes and a diverse set of exotics including synthetics-178, FIGS Drought-86, FIGS Heat-11, Mexican landraces-8, traditional breeding line-3, traditional Iranian variety-6, and Genebank line-3. Synthetics were derived from *Aegilops squarrosa* as well as *Aegilops seasrsii*. Therefore, the pre breeding germplasm set used in this study represented a genetically diverse material. Selection of the plant material not based on a previously available phenotype or genotype information is a way to have an unbiased estimate. The phenotypic experiment conducted in this study is independent of the previous report of Singh et al. [11].

### 4.2. Population Structure and Linkage Disequilibrium

Population structure in this panel has been investigated in a previous study by Ledesma-Ramírez et al. [12]. Briefly, three subpopulations were detected in the panel; subpopulation 1 was sharing the pedigrees of Baj#1, Reedling#1, Villa Juarez F2009, and Tacupeto F2001, subpopulation 2 was sharing the pedigrees of Seri.1b//Kauz/Hevo/3/Amad*2/4/Kiritati; and subpopulation 3 was sharing the pedigrees of Fret2*2/4/Sni/Trap#1/3/Kauz*2/Trap//Kauz/5/Kachu. Genome wide linkage disequilibrium (LD) decay was observed at 2.5 cM and, in subgenomes LD decay values were 0.25, 0.50, and 0.25 cM for A, B, and D genome, respectively [12].

### 4.3. Experimental Setup

The plant material was sown at the experimental station “campo experimental Norman E Borlaug (CENEB)” of CIMMYT Obregon, Sanora during the years 2015–2016 and 2017–2018. This diverse germplasm was planted in two replicates following the alpha lattice design under well-watered and drought conditions. Each genotype was planted in 2 rows each with 2-meter length and 40 cm distance between each row. Five irrigations, each containing approximately 125 mm, were given to normal irrigated treatment. In contrast, to ensure the drought stress and to avoid the border effect, plant material was grown 5-meter away from the water channel and from the border. Durum wheat was grown in the vacant area all around the experiment while only two irrigations (approximately 180 mm in total) were applied until the tillering stage, and drought stress was imposed at flowering by skipping the irrigations of pre and post anthesis stage (precipitation data is provided in Appendix A).

### 4.4. Phenotypic Characterization

Different agro-physiological parameters were recorded during and after maturity and are as follows:(a)Canopy temperature depression (CTD) was measured with infrared thermometer sixth sense (LT300). The measurement was adjusted 0.5 m distance above the canopy and viewing angle maintained around 45° middle area about 2 m of each plot was targeted.(b)Normalized difference in vegetative index (NDVI) was measured with the help of Green seeker variable rate application and mapping systems with portable sensor (Mod:505) for 5 s for both well-watered and drought genotypes. The average data were used with values 0–1 (no-green to maximum green).(c)Days to heading (DTH) were recorded by counting the number of days taken from sowing to 50% flowering.(d)Days to maturity (DTM) were counted by number of days taken from sowing to until all spikes were turn to their specific color of maturity.(e)Plant height (PH) was measured from 5 random plants of each replication from ground level to the end spike excluding awns(f)Spike length (SL) was calculated by taking 5 random spikes from each replication and length was measured from the start of spike to the end of spike excluding awns(g)Number of grains spike^−1^ (NGS) were counted by taking five random spikes from each replication and their averages were recorded(h)Thousand kernel weight (TKW) were recorded by counting thousand grains from each replication and their weight was recorded in grams(i)Kernel abortion (KA) was recorded using following formula:(NGS for drought plants/NGS for well-watered) × 100(j)Grain yield (Y) of each genotype was recorded after harvesting the both the rows and yield was expressed in kilograms per hectare.

### 4.5. Molecular Characterization

Leaf samples were taken at booting stage from TC_1_F_5_ plants, snap frozen in liquid nitrogen and stored at −80 °C until further analysis. Genomic DNA was extracted using modified cetyltrimethylammonium bromide (CTAB) protocol [11], followed by quantification by Nano-Drop 8000 spectrophotometer V 2.1.0 Thermo Fisher Scientific, Waltham, MA, United States. Likewise, for genotypic characterization of these samples, DArTseq™ technology (http://www.diversityarrays.com/dart-application-dartseq) of Genetic Analysis Service for Agriculture (SAGA) service unit at CIMMYT headquarters (Texcoco, Mexico) was used. High quality 58,378 SNP markers were generated and reported by Singh et al. [11]. The proportion of samples with genotypic score not recorded as missing data (call rate) and the proportion of technical replicate assay pairs for which the marker score was consistent (average reproducibility) were used to select the markers. The result was only 7180 markers used for the final analysis out of 12,071 SNPs. In addition, 100K-marker DArT-seq consensus map released by Diversity Arrays Technology Pty Ltd. (DArT, Canberra, Australia) (http://www.diversityarrays.com/sequence-maps) was used to define the chromosome location, marker order, and genetic distances. To explain the threshold significance level, a Bonferroni test was used. 

### 4.6. Haplotype Characterization

While constructing haplotype blocks, two approaches were taken: (a) haplotype blocks were searched within every 5 cM, and (b) haplotype blocks were searched with every 500 kb. The haplotype blocks were 99% similar by using both approaches. Haplotype blocks obtained by using genetic positions are only utilized in GWAS. Haplotype map was generated in R using the algorithm-based script described by Gabriel et al. [53]. In brief, D prime (D′) was generated through 95% confidence and three categories were made for the comparison called “strong linkage disequilibrium (LD)”, “inconclusive”, or “strong recombination”. The haplotype block was created using the R code as used in a previous study [11]. Briefly, we calculated linkage disequilibrium parameter D’ and D’ 95% confidence intervals between SNPs and each comparison was categorized as “strong LD,” “inconclusive,” or “strong recombination.” A haplotype block was created if 95% of the comparisons in one block were in “strong LD.” Cut off *p*-value for Hardy Weinberg was established at 0.001, while minimum value for marker allele frequency was set to 0.05. Haplotypes were not constructed for the individuals having more than 75% missing data. If the multiple SNPs were indicating the same genetic position, only the first marker was considered to construct haplotype map. The haplotypes were displayed as blocks of marker numbers and alleles and “HB” prefix was used for each haplotype block followed by the chromosome number followed by dot and then the increment number from 1 to N, where N is the total number of the haplotype blocks along the chromosome.

### 4.7. Genome Wide Association Study 

Population structure and genome wide linkage disequilibrium (LD) decay has been reported in this panel before by Ledesma-Ramírez et al. [12]. Principal component analysis (PCA) detailed in [12] was used while kinship matrix was generated using the VanRaden algorithm in GAPIT v 3.0. among the variables. A mixed linear model (MLM) was used with PCA as a fixed effect and kinship as a random effect. Significance of a marker was decided by two factors: (a) *p*-value (0.001 and 0.0001 corresponded to the bottom 0.1 percentile of yield and yield related traits) and (b) deviation of its *p*-value from the normal distribution curve. The following equations were used to determine GLM and MLM, respectively:Y_GLM_ = Aα + Bβ + e(1)
Y_MLM_ = Aα + Bβ + Cμ + e(2)
where vector of phenotype is represented by Y, vector of marker genotypes is represented by A; principal components matrix is represented by B; C is representing the relative kinship matrix; α, β, and μ are the corresponding effects; and residual effects are represented by e in the matrix. The A and B matrices were fitted as fixed effects, and C and e matrices were fitted as random effects.

## 5. Conclusions

Drought stress at flowering in bread wheat directly reduces grain number and grain weight. Our experimental results indicate that the transgressed genetic diversity from landraces and synthetic derivatives can improve Y under terminal drought through improvement in CTD and TKW. GWAS analysis coupled with haplotype-based association indicates stable association of certain markers for higher Y at chromosome 4A over both the years under drought conditions. These marker sequences have been studied in wheat before, but their function is not known. Likewise, SL is found to be associated at chromosome 2D both under irrigated and drought conditions, where HB6.10 was associated with higher SL. GWAS and haplotype-based association indicates that HB7.12 is reducing KA. Stable associations of certain markers were estimated at chromosome 3B to obtain higher TKW under irrigated conditions for both the years. These sequences have been reported in Chinese spring cultivar earlier, but their function was again unknow. Identified sequence tags can further be utilized in marker assisted selection through developing and validating corresponding KASP assays to improve the drought tolerance in wheat. This work is in progress at CIMMYT, Mexico at present. In addition, the germplasm materials performing better under terminal stage drought stress can be helpful to improve the tolerance of bread wheat commercial cultivars. 

## Figures and Tables

**Figure 1 ijms-21-03156-f001:**
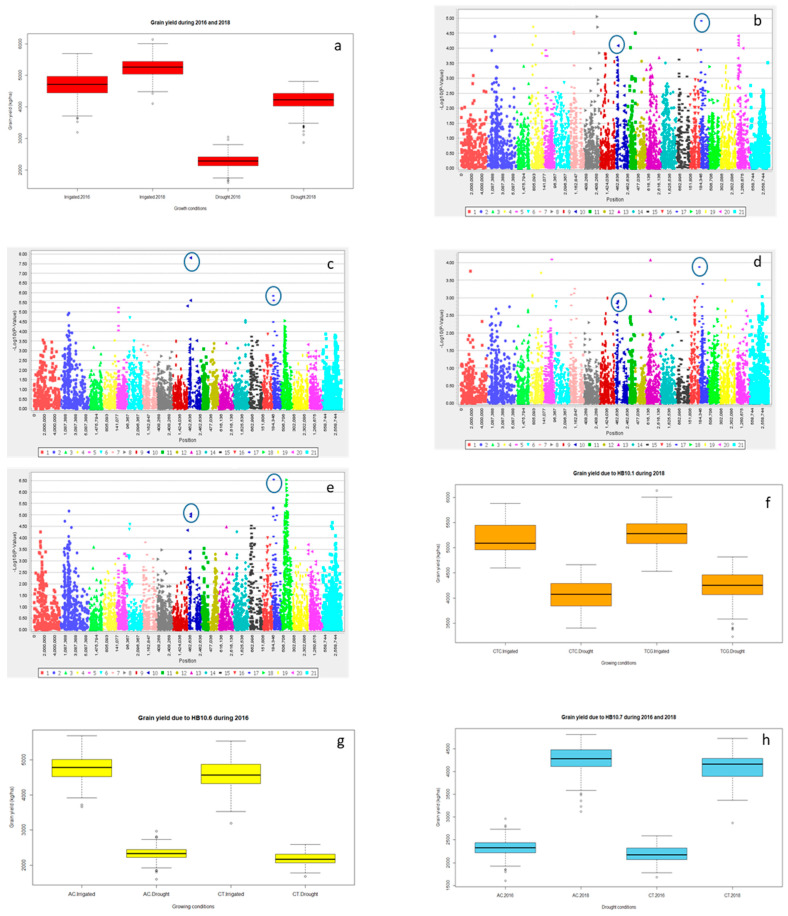
Boxplot indicating variation for grain yield under irrigated and drought conditions for the years 2016 and 2018 (**a**); marker trait association using GLM for irrigated (**b**) and drought (**c**) conditions 2016, and irrigated (**d**) and drought (**e**) conditions 2018. HB 10.1 showing differences in grain yield under irrigated and drought condition for 2018 (**f**), HB 10.6 showing differences in grain yield for irrigated and drought conditions in 2016 (**g**) and HB 10.7 showing differences for grain yield for drought conditions for 2016 and 2018 (**h**).

**Figure 2 ijms-21-03156-f002:**
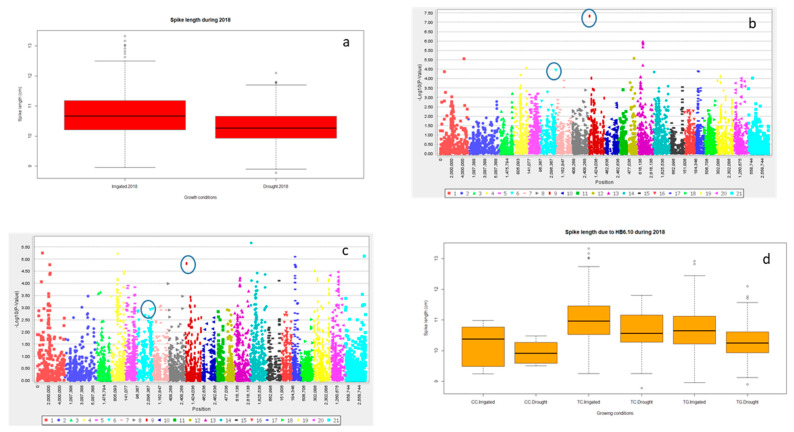
Boxplot showing variation for spike length under irrigated and drought condition for the year 2018 (**a**); marker trait association through GLM under irrigated (**b**) and drought (**c**) condition 2018, HB 6.10 showing differences in spike length (**d**).

**Figure 3 ijms-21-03156-f003:**
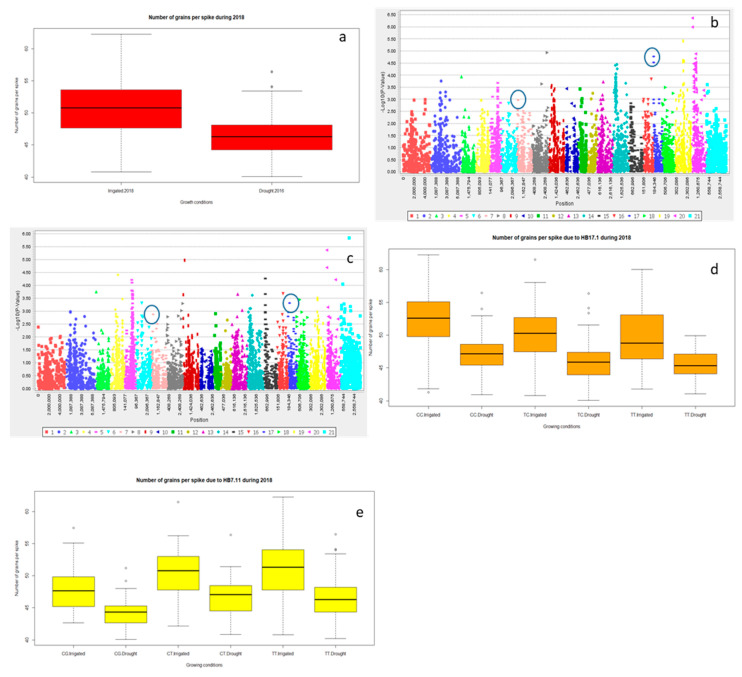
Boxplot showing variation in number of grains spike-1 under irrigated and drought conditions for 2018 (**a**); marker trait association through GLM under irrigated 2018 (**b**) and drought (**c**) conditions 2018. HB 17.1 showing differences in number of grains spike^−1^ (**d**) and HB 7.11 showing differences in number of grains spike^−1^ (**e**).

**Figure 4 ijms-21-03156-f004:**
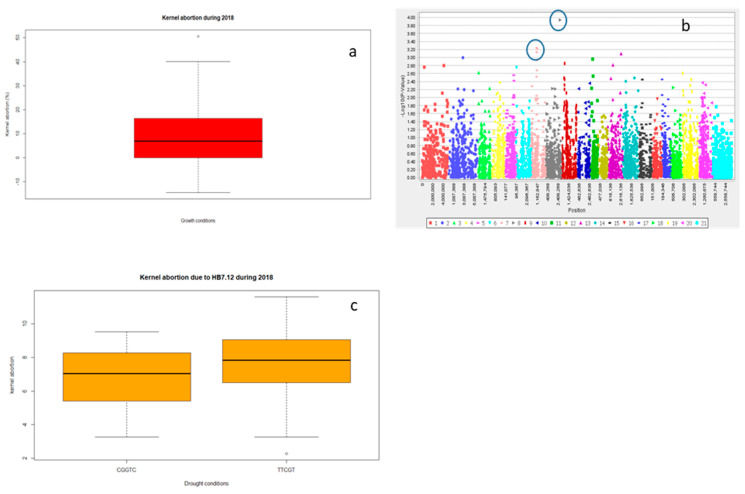
Boxplot showing variation in kernel abortion under drought during 2018 (**a**). marker trait association through GLM under drought 2018 (**b**) and haplotype block (HB) 7.12 showing differences in kernel abortion (**c**).

**Figure 5 ijms-21-03156-f005:**
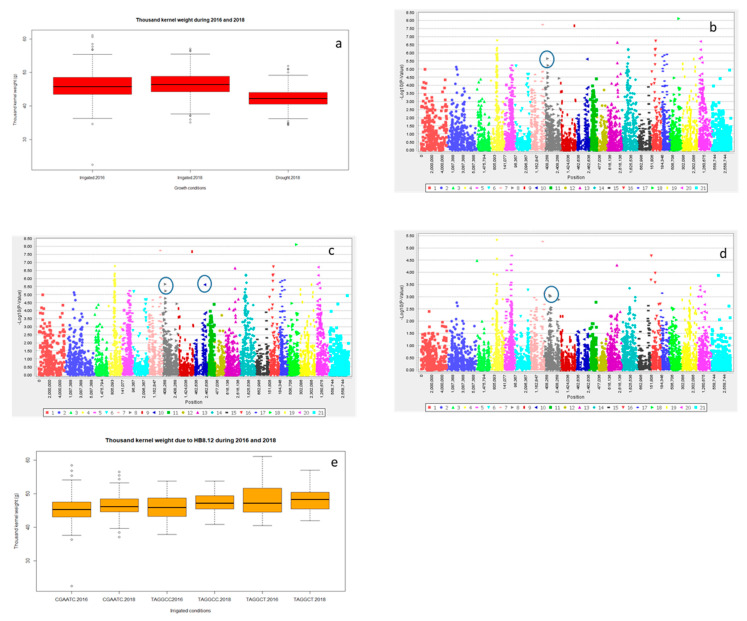
Boxplot showing variation for thousand kernel weight under irrigated conditions 2016 and irrigated and drought condition for 2018 (**a**). marker trait association through GLM under irrigated 2016 (**b**), irrigated 2018 (**c**) and drought (**d**) conditions 2018. HB 8.12 showing differences in thousand kernel weight under irrigated conditions for 2016 and 2018 (**e**).

**Figure 6 ijms-21-03156-f006:**
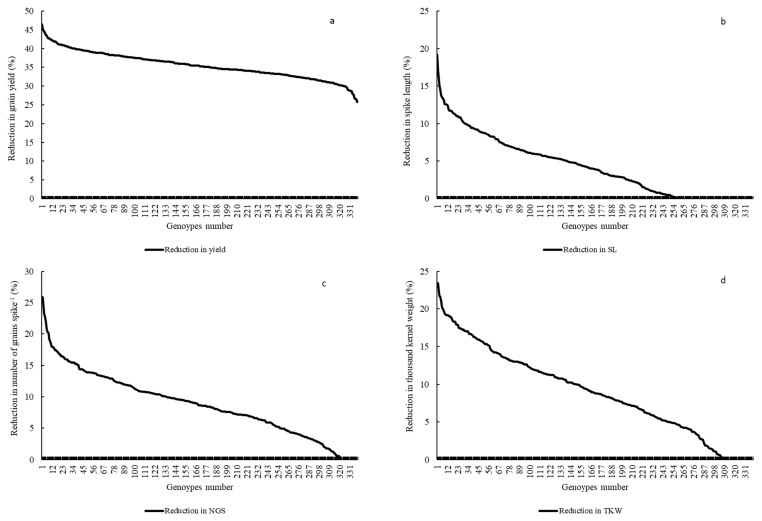
Graphs showing average reduction of different traits in 339 genotypes for grain yield (**a**), spike length (**b**), number of grains spike^−1^ (**c**), and thousand kernel weight (**d**).

**Figure 7 ijms-21-03156-f007:**
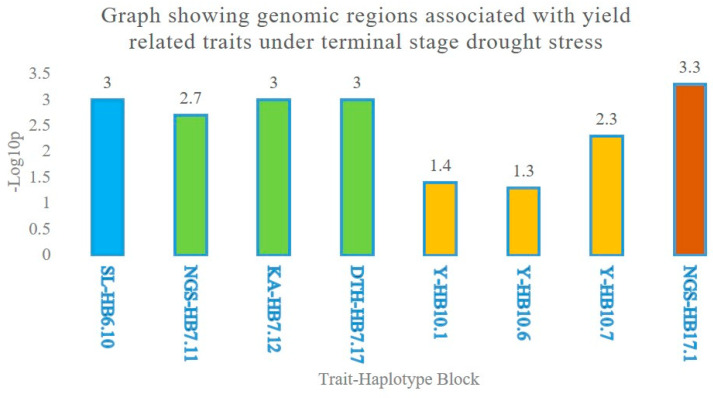
Graph showing genomic associations for grain yield and related traits under terminal stage drought stress in wheat. On *x*-axis traits and haplotype blocks were denoted and –Log10p value was plotted on the *y*-axis. Y = Grain yield, KA = Kernel abortion, SL = Spike length and NGS = Number of grains per spike.

**Table 1 ijms-21-03156-t001:** Mean squares, standard error, and significance level of different plant traits.

Variables	DTH	DTM	PH	NDVI	CTD	NGS	SL	TKW	Y	KA
2018-IR	67.41 ± 0.15	120.7 ± 0.1	96.7 ± 0.18	0.76 ± 0.0002	24.15 ± 0.004	50.82 ± 0.17	10.71 ± 0.03	46.44 ± 0.1	5242.59 ± 11.9	
2016-IR	77.09 ± 0.02	119.99 ± 0.1	99.7 ± 0.2					45.96 ± 0.2	4698.03 ± 15.3	
2018-DR	64.84 ± 0.01	104.77 ± 0.2	87.33 ± 0.13	0.65 ± 0.001	22.87 ± 0.006	46.24 ± 0.11	10.32 ± 0.02	42.30 ± 0.1	4203.02 ± 12.4	9.31 ± 0.4
2016-DR	76.08 ± 0.12	110.31 ± 0.1	81.95 ± 0.19						2277.12 ± 8.38	
Genotypes	ns	<0.001	<0.001	ns	ns	ns	<0.001	<0.001	<0.001	
Year	<0.001	<0.001	<0.001						<0.001	
Treatment	<0.001	<0.001	<0.001	<0.001	<0.001	<0.001	<0.001	<0.001	<0.001	<0.001
GenotypesxYear	ns	<0.001	<0.001						<0.05	
GenotypesxTreatment	ns	ns	ns	<0.05	ns	ns	ns	<0.001	ns	
YearxTreatment	<0.001	<0.001	<0.001						<0.001	
GenotypesxYearxTreatment	ns	ns	ns						ns	
Heritability broad sense (h_bs_) Irrigated	0.77	0.85	0.81	0.25	0.13	0.72	0.80	0.84	0.61	
Heritability broad sense (h_bs_) Drought	0.08	0.82	0.71	0.68	0.22	0.6	0.72	0.67	0.65	0.17

ns = non-significant, IR = Irrigated, DR = Drought; DTH = Days to heading, DTM = Days to maturity, PH = Plant height, NDVI = Normalized difference vegetation index, CTD = Canopy temperature depression, NGS = Number of grains, SL = Spike length, TKW = Thousand kernel weight, Y = Grain yield and KA = Kernel abortion.

**Table 2 ijms-21-03156-t002:** Mix linear model (MLM) describing the location of different traits on chromosomes.

Year	Treatment	Trait	Marker	Chromosome	Log10	*p*-Value	R^2^ (%)
2018	Irrigated	SL	1,216,938	3D	5.46	3.43 × 10^−6^	8.2
	Drought		1,216,938	3D	2.39	0.0041	2.8
2018	Irrigated	SL	1,113,306	2D	3.32	4.76 × 10^−4^	4.8
	Drought		1,046,144	2D	1.98	0.01047	2.1
2018	Irrigated	NGS	3,064,504	6B	2.46	0.00346	2.9
	Drought		2,256,902	6B	2.54	0.00287	2.9
2018	Irrigated	NGS	1,079,850	3A	2.12	0.00761	2.3
	Drought		1,190,017	3A	2.27	0.00532	2.6
2018	Drought	KA	4,910,194	3B	2.78	0.00167	3.4
	Drought	KA	1,076,524	3A	2.71	0.0019	3.1
2016	Irrigated	TKW	2,260,800	3B	2.04	0.0096	2.5
2018	Irrigated		3,064,863	3B	3.02	9.61 × 10^−4^	3.7
	Drought		988,433	3B	3.08	8.36 × 10^−4^	3.9
2018	Drought		1,664,941	4A	3.06	8.65 × 10^−4^	4.0
2016	Irrigated	Y	993,866	6B	1.99	0.01024	2.1
	Drought		1,219,677	6B	2.47	0.0034	2.9
2018	Irrigated		3,064,504	6B	3.28	5.22 × 10^−4^	4.1
	Drought		1,219,677	6B	4.57	2.70 × 10^−5^	6.1
2016	Irrigated	Y	1,101,137	4A	2.66	0.0022	3.6
	Drought		1,041,102	4A	3.40	3.94 × 10^−4^	5.0
2018	Irrigated		1,091,070	4A	2.64	0.0023	3.2
	Drought		1,041,102	4A	2.30	0.00502	2.4

SL = Spike length, NGS = Number of grains, KA = Kernel abortion, TKW = Thousand Kernel Weight and Y = grain yield.

**Table 3 ijms-21-03156-t003:** Association of haplotype blocks with grain yield and related traits under terminal stage drought stress in wheat.

Traits	HB	Clone ID	Position (cM)	Significance	R^2^ (%)	Chromosome
**Y**	HB10.1	1,054,658	11.64	*	6.9	4A
		1,718,021	10.76			4A
		2,248,753	13.13			4A
	HB10.6	1,130,105	28.98	*	3.0	4A
		1,217,569	28.84			4A
	HB10.7	3,024,473	31.63	**	5.9	4A
		979,934	32.41			4A
**TKW**	HB8.12	1,082,315	49.77	**	8.4	3B
		1,082,914	50.69			3B
		997,119	50.81			3B
		1,110,218	51.27			3B
		1,010,250	50.17			3B
		1,130,393	53.18			3B
**KA**	HB7.12	1,038,112	61.93	***	10.1	3A
		1,052,021	61.93			3A
		2,278,246	60.45			3A
		1,250,769	60.37			3A
		1,040,728	61.93			3A
**SL**	HB6.10	3,029,487	142.74	****	6.7	2D
		1,123,029	147.6			2D
**DTH**	HB7.17	1,150,200	77.13	***	5.5	3A
		3,024,325	76.17			3A
**DTM**	HB2.5	3,064,540	98.51	**	6.0	1B
		1,093,781	99.06			1B
**NGS**	HB17.1	3,222,182	2.64	****	7.4	6B
		1,050,615	6.34			6B
**NGS**	HB7.11	990,692	56.45	**	5.9	3A
		3,023,149	56.5			3A

Significance level presented as *: <0.05, **: <0.01,***: <0.001 and ****: <0.0001. Y = Grain yield, TKW = Thousand kernel weight, KA = Kernel Abortion, SL = Spike length, DTH = Days to heading, DTM = Days to maturity and NGS = Number of grains per spike. HB = Haplotype blocks, sequence tags corresponding to each HBs are presented in an adjacent column headed as clone ID.

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
