# Peer review of "Molecular Markers Associated with Agro-Physiological Traits under Terminal Drought Conditions in Bread Wheat"

_ijms, 2020, doi:10.3390/ijms21093156_

Round 1

Reviewer 1 Report

Dear Authors,

The manuscript titled “Genome wide association (GWA) analysis of wheat pre-breeding germplasm for terminal drought stress” deals within the scope of the International Journal of Molecular Sciences, by investigating an interesting topic of research and should be of interest to the readers.

However, some small mistakes must be corrected before the publication.

Here I report some suggestions:

In the attachment there are some suggestions:

Line 4 please mark the corresponding authors with "*"

Line 12 please note in the article the *correspondent author's email address, according to the requirements specified in the instructions for authors

L 94-98 please provide Supplementary Materials for Figure S2a; Figure S4a and Figure S5a; Figure S1a; Figure S3a…

Also please provide Supplementary Materials (Tables S1,2,3 and Figures S1,2,3,…) for all the text

L 100-102 – “Y was varying from 1653 kg ha-1 under drought condition to 5788 kg ha-1 under irrigated conditions.; “…..Y was observed in pre- breeding line (GID-7641264) where Y was reduced from 4437 kg ha-1 to 2556 kg ha-1”. — these data cannot be identified in the tables ... are they presented in the supplementary material? please refer to these

Please check and correct all similar issues in the whole paper. Some data cannot be identified in the tables / figures!!!

L 109-111 In the text, Figure 6c is indicated for both NGS and TKW; the figure seems to be the opposite, as noted. It is about figures 6c and 6d... please check

L 118 please correct the value 0.16 hbs for KA by 0.17 hbs. This is how it appears in the table 1… please check

L 124-137 For section 2.1. GWAS analysis in diverse pre-breeding germplasm (i. Physiological traits and i. Plant height and earliness please provide Supplementary Materials (Tables S and Figures S)

Line 155 please correct “… (Figure 3.f) ....” with “… (Figure 3.e).…”

Please increase the quality of figure 1 – 6

Line 171 please correct “… (Figure 3.e-f) ....” with “… (Figure 3.d-e).…”

Author Response

Response to reviewer 1

Dear Reviewer

Thank you very much for your comments 

I have incorporated the suggested changes which are as under;

  1. Line 4please mark the corresponding authors with "*"

Response: “*” for corresponding authors are added

  1. Line 12 please note in the article the *correspondent author's email address, according to the requirements specified in the instructions for authors

Response: Email addresses have been added as suggested

  1. L 94-98please provide Supplementary Materials for Figure S2a; Figure S4a and Figure S5a; Figure S1a; Figure S3a…Also please provide Supplementary Materials (Tables S1,2,3 and Figures S1,2,3,…) for all the text

Response: Supplementary figures are provided and added at the end of references

  1. L 100-102– “Y was varying from 1653 kg ha-1 under drought condition to 5788 kg ha-1 under irrigated conditions.; “…..Y was observed in pre- breeding line (GID-7641264) where Y was reduced from 4437 kg ha-1 to 2556 kg ha-1”. — these data cannot be identified in the tables ... are they presented in the supplementary material? please refer to these

Response: Mean squares and standard error is added in Table 1 while text has been improved accordingly. While text without tables or figures eliminated

  1. L 109-111In the text, Figure 6c is indicated for both NGS and TKW; the figure seems to be the opposite, as noted. It is about figures 6c and 6d... please check

Response: Many thanks and figures number are changed accordingly

  1. L 118please correct the value 0.16 hbs for KA by 0.17 hbs. This is how it appears in the table 1please check

Response: Value is changed accordingly in table

  1. L 124-137For section 2.1. GWAS analysis in diverse pre-breeding germplasm (i. Physiological traits and i. Plant height and earliness please provide Supplementary Materials (Tables S and Figures S)

Response: Supplementary Figures and Tables are provided at the end of the revised manuscript

  1. Line 155 please correct “… (Figure 3.f) ....” with “… (Figure 3.e).…”Please increase the quality of figure 1 – 6

Response: Figures number is improved accordingly while quality of the figures is also improved. These figures can also be provided as separate file to handling editor

  1. Line 171 please correct “… (Figure 3.e-f) ....” with “… (Figure 3.d-e).…”

Response: Figures number is improved accordingly

Reviewer 2 Report

The manuscript present GWAS studies in wheat for yield and yield components in control and drought conditions. Problem is significant and results should be of great interest and importance. Plant material is very interesting, genotyping is sufficient but I have serious concerns about phenotyping. It seems that phenotypic data were already published in part [11]. Unfortunately in the version of manuscript I have downloaded for evaluation no supplementary files were provided - and in fact there are missing data - that justify my negative recommendation. 

Quick search for regions reported in this manuscript indicate that some regions not reported previously [11] have been found. 

In results, no supplementary materials. line 98: "In contrast ...." is not clear. Figures should be rearranged it is better to put components 1a to 5a in single fig. Figures are not very informative for me - to confirm significance of difference (from table 1) it would be necessary to show confidence intervals. Lines 126-137 are based on not available supplementary materials. On Manhattan plots there are several regions omitted - what significance threshold was used to select haplotypes - in present form selection of regions seems subjective. In results authors must define differences between already identified effects (HB10.7) and new effects. Manuscript should be suitable for publishing when new added results will be of interest. It may be inconsistency - why LD is calculated for genetic distances and haplotypes are defined for physical ones?

I am not able to evaluate in silico analysis without supplementary data but, some regions (line 250) are huge. How many genes was predicted in this region? Step by step details of methodology of in silico analysis should be provided. In the methods, experimental material is selection of material previously reported in [11] - but what was left and what was eliminated?

Information on data published previously (phenotype) should be provided. More details on drought treatment seems necessary - some rainfall information for period after tilling should be helpful.

In molecular characterization - is it OK that number of genotypes was reduced and number of polymorphic markers remained as already reported in [11]? Details on threshold used to reject markers are missing (line 391-3). 

Author Response

Response to Reviewer 2

Dear Reviewer

Thank you very much for your comments 

I have incorporated the suggested changes which are as under;

  1. The manuscript present GWAS studies in wheat for yield and yield components in control and drought conditions. Problem is significant and results should be of great interest and importance. Plant material is very interesting, genotyping is sufficient but I have serious concerns about phenotyping. It seems that phenotypic data were already published in part [11]. Unfortunately in the version of manuscript I have downloaded for evaluation no supplementary files were provided - and in fact there are missing data - that justify my negative recommendation.

Response: Supplementary files and Tables are added at the end of manuscript. 

  1. Quick search for regions reported in this manuscript indicate that some regions not reported previously [11] have been found. In results, no supplementary materials. line 98: "In contrast ...." is not clear. Figures should be rearranged it is better to put components 1a to 5a in single fig. Figures are not very informative for me - to confirm significance of difference (from table 1) it would be necessary to show confidence intervals.

Response: Figures have been improved and incorporated again. Table 1 is improved by inclusion of mean squares and standard error

  1. Lines 126-137 are based on not available supplementary materials. On Manhattan plots there are several regions omitted - what significance threshold was used to select haplotypes - in present form selection of regions seems subjective. In results authors must define differences between already identified effects (HB10.7) and new effects. Manuscript should be suitable for publishing when new added results will be of interest.

Response: In Discussion section, new effects of the regions have been explained. See for example, on page 13 second paragraph the below lines

“Also, association of this genomic region with CTD and NDVI indicates that the Y increase under drought was likely due to biomass increase under drought stress environments” Previously, the physiological basis of yield enhancement was not known but due to the present study many underlying physiological mechanisms explaining drought tolerance. Moreover, Kernel abortion was never investigated before on this panel. This is the first report of investigation of kernel abortion trait under drought stress.

  1. It may be inconsistency - why LD is calculated for genetic distances and haplotypes are defined for physical ones?

Response: Sorry for the confusion. While constructing haplotype blocks two approaches were taken; a) in the first haplotype blocks were searched within every 5cM and b) in the second approach haplotype blocks were searched with every 500kb. The haplotype blocks were 99% similar by using both approaches. Haplotype blocks obtained by using genetic positions are only utilized in GWAS. This has been clarified now in revised version.

  1. I am not able to evaluate in silico analysis without supplementary data but, some regions (line 250) are huge. How many genes was predicted in this region?

Response: Supplementary data has been provided and although the region is huge in lines 250 but our in-silico analysis indicate only one gene which is coding for heat shock protein

  1. Step by step details of methodology of in silico analysis should be provided.

Response: Precisely, to find the candidate genes, the physical starting point of all the markers in the associated haplotype blocks along with chromosome name was put in Ensembl Plants database (https://plants.ensembl.org/Triticum_aestivum/Info/Index).  Since, GBS sequences are shorter (about 64 to 69 bp), 300 bp sequences were added before and after the SNP and used for BLAST in Ensemble server. For some markers, many SNPs of the haplotype block fell within gene sequences and hence were classified as direct gene hits (Table S4). However, for markers where none of the SNPs of the block fell within a gene, potential candidate genes were picked 2 Mb upstream and downstream of the SNPs which had known role in abiotic stress tolerance. Sometimes annotations were not available in Triticum aestivum genome and for such cases orthologous genes in related species such as Aegilos tauschii or model crops such as Brachypodium, Oryza sativa were screened with known predicted functions using the comparative genomics tool in Ensembl.

  1. Information on data published previously (phenotype) should be provided. More details on drought treatment seems necessary - some rainfall information for period after tilling should be helpful.

Response: Supplementary data about rainfall during stress period is provided in figure S6

  1. In molecular characterization - is it OK that number of genotypes was reduced and number of polymorphic markers remained as already reported in [11]? Details on threshold used to reject markers are missing (line 391-3). 

Response: Briefly, we calculated linkage disequilibrium parameter D’ and D’ 95% confidence intervals between SNPs and each comparison was categorized as “strong LD,” “inconclusive,” or “strong recombination.” A haplotype block was created if 95% of the comparisons in one block were in “strong LD.”  Cut off p-value for Hardy Weinberg was established at 0.001 while minimum value for marker allele frequency was set to 0.05. Haplotypes were not constructed for the individuals having more than 75% missing data. If the multiple SNPs were indicating the same genetic position, only the first marker was considered to construct haplotype map. The haplotypes were displayed as blocks of marker numbers and alleles and “HB” prefix was used for each haplotype block followed by the chromosome number followed by dot and then the increment number from 1 to N where, N is the total number of the haplotype blocks along the chromosome.

Reviewer 3 Report

The reviewed manuscript presents the results of a genome-wide search for SNPs and haplotypes potentially associated with reaction to drought stress at terminal stage of wheat growth. The study is based on the analysis of ten agro-physiological traits in a representative set of 339 pre-breeding wheat lines grown during two seasons under contrasting well-watered and drought conditions. The manuscript contains two tables and five informative figures. Bibliography list includes 63 references and properly covers relevant literature in the field of research. In general, the research is well designed. The study was performed with double repetition and on the base of original genotypes which have been created based on the three-way top crosses. The results undoubtedly have scientific significance and can to be published, however, manuscript require some editing and technical corrections.

  1. It is necessary to improve quality of the figures: the designations on them for the most part are unclear and hard to see.
  2. The text needs technical and editing improvement. Sometimes the authors are free to use special scientific terms. For example: the expressions like “genomic regions associated with the terminal stage drought stress in wheat”, “the location of different traits on chromosomes”, “significant association of NGS and Y at chromosome 3A”, “allele TC was linked with higher SL” are incorrect and difficult to understand.
  3. The article title “Genome wide association (GWA) analysis of wheat” pre-breeding germplasm for terminal drought stress” is incorrect. Genome wide association studies aim to revealing genome locations of genetic factors responsible for certain traits. Therefore, the purpose of the study was to identify location of markers associated with wheat agro-physiological traits variation under terminal drought conditions (but not associated with drought stress).
  4. The key words list partly duplicate the words used in the title (that is undesirable) and should be revised.

Author Response

Response to Reviewer 3

Dear Reviewer

Thank you very much for your comments 

I have incorporated the suggested changes which are as under;

  1. It is necessary to improve quality of the figures: the designations on them for the most part is unclear and hard to see.

Response 1: Figures designation and quality are improved in the revised version of the article. These figures are also be provided as separate file to handling editor

  1. “stress in wheat”, “the location of different traits on chromosomes”, “significant association of NGS and Y at chromosome 3A

Response 2: Sentence “stress in wheat” from the abstract is changed with “identification of genomic regions associated with agro-physiological traits at terminal stage drought stress in wheat” sentence “location of different traits on chromosome” from the tables” is changed with “association of yield and yield related”.

Similarly, sentence “significant association of NGS and Y at chromosome 3A” is changed according to results

  1. The text needs technical and editing improvement. Sometimes the authors are free to use special scientific terms. For example: the expressions like “genomic regions associated with the terminal stage drought”, “allele TC was linked with higher SL” are incorrect and difficult to understand.

Response: “allele TC was linked with higher SL” is changed with “where allele TC was favorable allele and it was linked with higher SL in comparison to other alleles”. Quality of figures is improved in the revised article as advised.

  1. The article title “Genome wide association (GWA) analysis of wheat” pre-breeding germplasm for terminal drought stress” is incorrect. Genome wide association studies aim to revealing genome locations of genetic factors responsible for certain traits. Therefore, the purpose of the study was to identify location of markers associated with wheat agro-physiological traits variation under terminal drought conditions (but not associated with drought stress).

Response: Title of manuscript is changed to “Molecular markers associated with agro-physiological traits under terminal drought conditions in bread wheat”

  1. The key words list partly duplicate the words used in the title (that is undesirable) and should be revised.

Response: Keywords are changed

Note: Results have been improved

Round 2

Reviewer 2 Report

In supplemented and revised version of the manuscript, some of my concerns were fixed but still some were omitted. The most important neutral criteria for selection significant effects were not applied. It is apparent from figs S1 and S2 that for declared p-value there are more significant effects than these presented in Table S1. (For irrigated NDVI at LOD 3 = p-val 0.001 there are hits for chromosome 2, 15, 18 and 19, for drought 1, 2, 4, 7, 13). For NDVI, reported (line 158) effect on 4A is below threshold.  In table S1, information on Log10 and derivative p-value is presented  (one of them p-val or Log is sufficient) instead of position of marker on chromosome - that is necessary to delimit QTL for subsequent candidate gene identification. The same unjustified selection of effects is for Figs S2-5 - this opens way to publish the same results for various effects in various journals - and reduces value of component reports. Therefore, I cannot reccomend this manuscript for publication. Moreover, kernel abortion is reported for single year, heritability is not calculated - and normally field results for single year are not acceptable for GWAS. 

In table S4 "S.#" -?, target SNP should be marked on AlleleSequence.

Candidate genes search in my opinion may need additional in-depth revision. Thank you for methodology. When effect is found for single marker proposed window of +/- 2Mb is OK, but annotation of DArT marker as gene-hit not release from the duty to search adjacent region for identification and reporting all qualified candidate genes. Possibly some KASPs can be proposed for extended DArTs. 

Author Response

Response to comments of reviewer2

Comment: In supplemented and revised version of the manuscript, some of my concerns were fixed but still some were omitted. The most important neutral criteria for selection significant effects were not applied. It is apparent from figs S1 and S2 that for declared p-value there are more significant effects than these presented in Table S1. (For irrigated NDVI at LOD 3 = p-val 0.001 there are hits for chromosome 2, 15, 18 and 19, for drought 1, 2, 4, 7, 13). In table S1, information on Log10 and derivative p-value is presented (one of them p-val or Log is sufficient) instead of position of marker on chromosome - that is necessary to delimit QTL for subsequent candidate gene identification. The same unjustified selection of effects is for Figs S2-5 - this opens way to publish the same results for various effects in various journals - and reduces value of component reports. Therefore, I cannot reccomend this manuscript for publication.

Response1: We fully agree with the reviewer regarding significance criteria. We have followed bonferroni criteria for determining the significance thresholds.

  • While presenting results we preferred haplotype blocks over SNPs,
  • We preferred multiple haplotype blocks in a genomic region rather one block for a particular trait.
  • We have revised the tables and figures accordingly.
  • Authors request reviewer to refer for newly added Table 3 and Figure7 to present robust results of haplotype block-trait analysis. Data of supplementary figures and table is revised according to the LOD 3 = p-val 0.001.

Revision in the text is as below

“Individually GLM and MLM reveled number of associations however, associations explained by both were further taken to explain the traits. GLM and MLM analysis revealed significant (P<0.001) associations of NDVI on different chromosomes however, common associations at the same chromosome were selected for irrigated and drought conditions where, significant and common associations were found on chromosome 1B and 3A for year 2018 (Figure S1.b-c and Table S1). For CTD, associations were calculated on chromosome 4B and 7B for irrigated and drought conditions respectively (Figure S2.b-c and Table S1). Association of CTD was also significant at 4A chromosome under drought condition. 

GLM and MLM of GWAS indicated significant association (P<0.001) of PH at chromosome 1B and 2A under irrigated and drought conditions for the year 2016 respectively. While, common association was found at chromosome 7A for the year 2018 (Figure S3.b-e and Table S1). No common association for DTH was recorded however, associations on chromosomes 3A, 2A, 1B and 7A were recorded for irrigated and drought conditions of 2016 and; irrigated and drought condition of 2018 respectively (Figure S4.b-e and Table S1). Moreover, on chromosome 3A, favorable allele TA of HB7.17 showed earliness for DTH in comparison to CC allele under irrigated conditions (Figure S4.f). Like DTH, no common association for DTM was recorded under irrigated conditions where, association on chromosome 5B and 5A was recorded for the year 2016 and 2018 respectively.  However, association on same chromosome (6A) was recorded under drought conditions for both the years (Figure S5.b-e and Table S1). Furthermore, HB2.5 on chromosome 1B with favorable allele AT showed earliness for DTM in comparison to other alleles (Figure S5.f).”

Comment2: For NDVI, reported (line 158) effect on 4A is below threshold. 

Response2: Statement was improved according to LOD value as follow;

Though with low significance level, association of three HBs for Y in same genomic region clearly indicates its importance for Y under terminal sage drought (Table 3, Figure 7). Previous report of Singh et al. [11] also emphasized on importance of this region on Y under drought. Population used in this study was a randomly selected subset of what was used by Singh et al. [11]. Results on chromosome 4A in this study are in concomitance with the previous study. Significance level for HB association with Y was low in this study as compared to what was in Singh et al. [11], most likely due to change in the population structure. Also, level of drought stress in 2016 was not as high to reduce yields under stress by 30% even. These are the most probable reasons for low level of significance. Association of three chromosome 4A HBs (in which two are consecutive) in the same genomic region for Y under drought stress and concomitance of results with that of Singh et al. [11] clearly signifies the relevance of this genomic region for Y under different moisture regimes.

Interestingly, association of this genomic region with CTD indicates that the Y increase under drought was likely due to cooler canopies under drought stress environments. Results therefore suggested that 4A genomic region is drought responsive”.

Comment3: Moreover, kernel abortion is reported for single year, heritability is not calculated - and normally field results for single year are not acceptable for GWAS. 

Response4: Kernel abortion was not recorded in two years due to feasibility. We agree with reviewers’ comment as don’t have two years data for kernel abortion (KA). So, heritability was calculated for available one-year data. Reports for GWAS with KA in wheat under drought are rare, therefore, a preliminary finding of this study has been presented. Also, KA showed a strong association with grain yield related parameters including- grain number and grain weight. Inclusion of this parameter provided us an indication toward cause of grain yield reduction under terminal stage drought stress in this study.

Comment5: In table S4 "S.#" -?, target SNP should be marked on AlleleSequence.

Response5: Table S4 is removed and necessary information is incorporated in Table 3 and Figure 7

Comment6: Candidate genes search in my opinion may need additional in-depth revision. Thank you for methodology. When effect is found for single marker proposed window of +/- 2Mb is OK, but annotation of DArT marker as gene-hit not release from the duty to search adjacent region for identification and reporting all qualified candidate genes. Possibly some KASPs can be proposed for extended DArTs. 

Response: The DArT-seq markers are like GBS tags of 66 base pairs usually. These markers provide multiple hits in genome and therefore can’t provide robust candidate gene search as compared to KASPs. We therefore removed the candidate gene search portion and suggested for development of KASP assays for the identified genomic regions, as suggested by reviewer.

Round 3

Reviewer 2 Report

Thank you for all explanations. I have no further remarks.